# Selected Genetic Factors Associated with Primary Ovarian Insufficiency

**DOI:** 10.3390/ijms24054423

**Published:** 2023-02-23

**Authors:** Mengchi Chen, Haotian Jiang, Chunping Zhang

**Affiliations:** 1Queen Mary School, Nanchang University, Nanchang 330006, China; 2Department of Cell Biology, College of Medicine, Nanchang University, Nanchang 330006, China

**Keywords:** POI, genetics, mutations, folliculogenesis, ovary

## Abstract

Primary ovarian insufficiency (POI) is a heterogeneous disease resulting from non-functional ovaries in women before the age of 40. It is characterized by primary amenorrhea or secondary amenorrhea. As regards its etiology, although many POI cases are idiopathic, menopausal age is a heritable trait and genetic factors play an important role in all POI cases with known causes, accounting for approximately 20% to 25% of cases. This paper reviews the selected genetic causes implicated in POI and examines their pathogenic mechanisms to show the crucial role of genetic effects on POI. The genetic factors that can be found in POI cases include chromosomal abnormalities (e.g., X chromosomal aneuploidies, structural X chromosomal abnormalities, X-autosome translocations, and autosomal variations), single gene mutations (e.g., newborn ovary homeobox gene (NOBOX), folliculogenesis specific bHLH transcription factor (FIGLA), follicle-stimulating hormone receptor (FSHR), forkhead box L2 (FOXL2), bone morphogenetic protein 15 (BMP15), etc., as well as defects in mitochondrial functions and non-coding RNAs (small ncRNAs and long ncRNAs). These findings are beneficial for doctors to diagnose idiopathic POI cases and predict the risk of POI in women.

## 1. Introduction

Female infertility refers to the inability to conceive after 6 months (for women over the age of 35) to 1 year of regular and unprotected sex. According to the Office on Women’s Health (OWH) of America, premature ovarian insufficiency (POI) is one of the most common causes of female infertility. POI, which is also known as primary ovarian insufficiency or premature ovarian failure, refers to female amenorrhea before the age of 40 due to non-functional ovaries caused by follicle atresia and the rapid loss of germ cells [1]. POI is manifested as primary or secondary amenorrhea with hormonal changes, such as increased gonadotropin levels (FSH > 25 IU/L) and decreased estradiol and anti-Müllerian hormone levels [1,2]. Moreover, there are many clinical presentations in POI women. Hot flashes, night sweats, and insomnia are all classic symptoms of POI, which coincide with reduced estrogen condition [3]. The increase in the POI cases, among which there is a vast number of POI women with unclear genetic diagnoses, justifies investigating the etiology of POI, which may be critical in the early diagnosis, treatment, and prevention [3].

Although POI is heterogenous and the causes of many cases remain unclear, various types of etiologies, such as genetic, autoimmune, iatrogenic, infectious, environmental, chemotherapeutic, and radiotherapeutic causes, have been determined [4,5,6]. Previous research established a relationship between the genetic effects and POI by studying POI in families (the prevalence of familial POI ranges from 4% to 31%) [7,8,9]. Moreover, there is an increasing prevalence of POI in adolescents. In a recent study, the research group gathered information on women under 21 years of age diagnosed with POI in 2000–2016 from all pediatric endocrinology units in Israel [10]. Among the 130 women with POI, the most common cause was Turner syndrome/mosaicism, accounting for 43% of cases. For non-Turner POI cases in this group, a significant increase in the incidence of POI was observed. This is due to new and more effective gene technology and the frequent occurrence of autoimmune diseases. The incidence rate of new POI diagnoses per 100,000 person-years increased year-by-year, especially in 2009–2016, indicating the remarkable incidence rate of POI in adolescents. Furthermore, the overall prevalence of genetic-associated POI is approximately 20–25% [11]. Therefore, taking genetic factors into account often leads to a more straightforward POI diagnosis. Previous reviews on the genetic factors of POI often focus on one or more aspects, such as the POI-related genes involved in meiosis or DNA repair, or they reveal a certain type of chromosomal mutation associated with POI, such as X-autosome translocations. In addition, the broader reviews only pay attention to the single gene mutations of POI and so on. In this review, we reasonably classify the genetic factors and non-syndromic POI-related genes (according to the biological process in which genes participate), based on previous research, so that the genetic factors and corresponding mechanisms of POI are more comprehensively and carefully summarized. We divided genetic causes of POI into four categories after reviewing and analyzing the content of other previous literatures: chromosomal abnormalities, single gene variants in non-syndromic and syndromic POI, mitochondrial dysfunction, and abnormal levels of non-coding RNAs (Figure 1). Thus, the present review provides essential information to help us better understand POI caused by genetic factors. We hope that this will act to improve the efficacy of diagnosis and treatment for POI patients.

## 2. Methods

We entered the following keywords “POI and genetic factors” (219), “POI and gene mutation” (217), “POI and variants” (209), “POI and a certain gene name”, “POI and mitochondria” (18), and “POI and non-coding RNA” (54) in Pubmed to search the articles and reviews on POI-related genetic factors in the past ten years. Overall, we searched for more than 717 articles. For the studies to be included in this review, the selected publications had to focus on the following: identifying the POI genetic factors in different POI populations via chromosomal analysis, candidate gene screening, genome wide association study, and various genome sequencing strategies. In addition, the role of POI candidate genes in animal models and studies on mitochondrial genes and non-coding RNAs associated with POI were also included. However, the literature identified was restricted to English language.

## 3. Chromosomal Abnormalities

Chromosomal abnormality is defined as a variation resulting from aneuploidy or structural defects in chromosomes. This can lead to many harmful and even lethal human genetic diseases, such as trisomies 21, 18, and 13 and sex chromosomes rearrangements. On this basis, researchers attempted to detect chromosomal variations using prenatal testing, which can be used to effectively avoid human genetic diseases caused by chromosomal abnormalities [12,13]. In addition, recent studies have shown that chromosomal abnormalities are responsible for POI [1,6,14]. The prevalence of POI caused by chromosomal abnormalities varies in different populations, with the values ranging from approximately 10% to 13% [4,15,16]. Chromosomal disorders cause POI via the depletion of primordial oocytes during early female development [1]. However, the mechanism involved in the loss of oocytes is not clearly understood. Moreover, defects in both the X chromosome and autosome can contribute to POI. Specifically, all X monosomies and trisomies, X chromosome deletions, X-autosome translocations, and autosomal translocations represent chromosomal abnormalities that can lead to POI.

### 3.1. X Chromosome Aneuploidies

Turner syndrome is a critical sex chromosomal disease in females with a 1 in 2500 incidence. It is caused by the complete or partial deletion of one sex chromosome [17,18]. TS has many hallmarks, including ovarian failure, and it is the most common genetic cause of POI, accounting for 4–5% of all POI cases [7,11,19]. Primordial follicle atresia and a reduction in the ovarian reserve are the circumstances under which TS causes POI, but its mechanism is unknown. A recent review suggested that TS-related POI was associated with the function and length of telomeres and epigenetic modifications [20]. Moreover, many scientists have shown that ovary-related genes were also responsible for the ovarian phenotype in TS patients [21]. The severity of TS patients’ symptoms depends on whether their genotype is 45X or mosaicism, such as 45X/46XX. The TS patients with a mosaic genotype are characterized by secondary amenorrhea, which means that this group of patients is fertile (producing a lower level of follicles) at first. However, fertility reduces over time, and eventually, POI develops [22,23]. In contrast, TS patients who lose a complete X chromosome are characterized by primary amenorrhea with a small chance of menarche [11].

Trisomy X syndrome (TXS) with a karyotype 47XXX is another sex chromosome aneuploidy that contributes to POI [24]. A relatively small body of literature is concerned with the connection between X trisomy and POI. In 2020, Shanlee et al. performed a case-control study and demonstrated that the level of anti-mullerian hormone (AMH) in TXS patients was lower than in healthy females, indicating that TXS females had a higher risk of suffering from POI [25]. In addition, in females with TXS, increasing follicle-stimulating hormone (FSH) and luteinizing hormone (LH) can cause menstrual cycle disturbance, which is associated with POI [26]. A previous survey also reported a patient with both blepharophimosis-ptosis-epicanthus syndrome (BPES) and TXS presenting with POI. However, this patient had a normal level of gonadotropins, which is rare in POI cases [27].

### 3.2. Structural X Chromosomal Abnormalities and X-Autosome Translocations

A number of studies have established a relationship between X chromosomal structural disorders (mainly X chromosomal deletions), X-autosomal translocations, and POI [11,26,28]. Furthermore, a recent study posited the existence of POI critical regions 1 and 2 on the X chromosome, which define the positions of the breakpoints for X chromosomal deletions and X-autosomal translocations related to ovarian functions, respectively [22]. The POI1 region consists of a part of the long arm of the X chromosome ranging from Xq24 to Xq27, but does not conclude the fragile X mental retardation 1 (FMR1) gene, while the Xq13.1 to Xq21.33 region belongs to the POI2 region. Although a considerable body of research has demonstrated the pathogenic role of structural X chromosomal abnormalities in POI, much less attention has been paid to this area over the past decade. By comparing and analyzing the statistics from four studies, the reasonable prevalence of X chromosome structural anomalies and X-autosome translocations related to POI was calculated to range from 4.2% to 12.0% [29,30,31,32]. Moreover, many POI candidate genes on the X chromosome can be found by analyzing the X-autosome translocations, some of which are introduced below.

### 3.3. Autosomal Abnormalities

Autosomal translocations, microdeletions, specific gene mutations, epistasis, and epigenetics associated with autosomal genes are all responsible for POI [33,34]. As has been previously reported, for chromosome variations, most POI cases are caused by X chromosome variations, while autosomal variations account for only a small number of POI cases [16,29,35]. The majority of previous and current studies focus on autosomal translocations and gene variants. According to a recent literature review, only 23 cases exhibited autosomal abnormalities (Robertsonian translocation, reciprocal translocation, and chromosomal inversion) associated with POI, which were identified in different POI populations with different ethnicities, including Chinese, Thai, and American [36]. Many autosomal genes associated with ovarian functions are discussed in the next section.

## 4. Single Gene Variants and Non-Syndromic POI

Aside from chromosomal abnormalities, single gene variations can also cause POI. The classical candidate gene approach is based on genes with known functions and experimental models in mice. In these, scientists establish a hypothesis regarding the connection between candidate genes and POI. Using this method, many genes, such as *BMP15*, *NOBOX*, and *FMR1*, have been discovered [37]. However, the range of candidate genes is restricted in this traditional approach. The appearance of many new strategies, such as genome-wide association studies (GWAS), can overcome the shortcomings of the traditional method. However, as a result of the low prevalence and high heterogeneity of POI, it is challenging to perform replicated experiments to prove the causation of these candidate genes using GWAS. Therefore, the majority of recent studies that investigate POI-related genes utilize another method, known as whole exome sequencing (WES). Moreover, next-generation sequencing (NGS) is also responsible for identifying variants. The associated genes identified in the last 10 years are classified according to the biological processes they participate in (Table 1), which are introduced below.

### 4.1. Meiosis and DNA Replication and Repair

A normal oocyte reserve is essential for females of reproductive age to give birth to a healthy baby. However, if any error occurs during meiosis, DNA replication, or DNA repair, the genetic information is negatively affected, leading to germ cell apoptosis and infertility. Therefore, collecting and investigating the genes involved in the critical processes of meiosis, DNA replication, and DNA repair is beneficial for obtaining a better understanding of POI.

#### 4.1.1. Helicase for Meiosis 1 (HFM1)

*HFM1* exists on chromosome 1p22 and encodes DNA helicase, which is only expressed in the ovary and testis. There are numerous studies that demonstrate the connection between *HFM1* and POI. A recent study conducted WES in a Chinese POI cohort containing two POI patients (the proband and her mother) and found a novel missense mutation of *HFM1* (c.3470G> A), which can affect mRNA transcription [88]. However, its role in protein levels needs further investigation. Moreover, one out of twenty-four POI patients in a cohort harbored two disease-causing variants in *HFM1* (c.3100G > A and c.1006 + 1G > T) [41]. Similar to c.3470G> A, c.1006 + 1G > T also disrupts RNA splicing. In addition, another mutation, c.3100G > A, alters the amino acid of the protein (p.G1034S). The way in which mutant *HFM1* causes POI is its ability to damage meiosis in the oocyte. However, the definite role of the *HFM1* gene in meiosis remains uncertain. The majority of associated studies have shown that the *HFM1* gene is involved in homologous recombination (HR) and synapsis [38,39,40]. Moreover, in 2020, Wang et al. was first to demonstrate that HFM1 gathers at the spindle pole and is responsible for normal spindle formation and function during meiosis in female mice oocytes via maintaining the usual activities of GM130 and p-Mapk proteins [89].

#### 4.1.2. DNA Primase Subunit 1 (PRIM1)

*PRIM1* is a protein-coding gene and is located at 12q13.3. It is one of the subunits of DNA primase, which is essential for DNA replication by synthesizing RNA oligonucleotide primers, so as to promote the production of new lagging and leading strands through DNA polymerase [42]. In addition, *PRIM1* is also related to DNA repair [90]. As previously reported in a meta-analysis, many loci in the corresponding genes were identified using GWAS that are associated with natural menopause age in European women. In this, *PRIM1* (SNP rs2277339) was the second strongest candidate gene [43]. Perry et al. proved that non-synonymous SNP rs2277339 in *PRIM1* was responsible for POI [91]. Another meta-analysis also recognized the relationship between *PRIM1* (SNP rs2277339) and natural menopause age in African American women [92]. However, the further consequences of mutations in SNP rs2277339 remain unclear. However, in 2016, Wang et al. published a paper concluding that perturbations in the coding region of *PRIM1* were not common among Chinese POI patients [93].

#### 4.1.3. Stromal Antigen 3 (STAG3)

*STAG3* (7q22.1) is also an important candidate gene in POI. Much of the current literature on gene mutations that cause POI pays particular attention to the effects of *STAG3* variants. Over the last 5 years, scientists have conducted WES in consanguineous families [94,95,96] and showed various novel homozygous pathogenic *STAG3* variants that cause POI, which included a missense variant (NM_012447.3:c.962G > A), two novel in-frame variants (c.877_885del, p.293_295del; c.891_893dupTGA, p.297_298insAsp), and a donor splice site variant (NM_012447.2: c.1573 + 5G > A). In addition, a series of reports focusing on different ethnicities (Senegalese, white British, and Brazilian POI patients) also revealed many POI-related pathogenic mutations in *STAG3* (c.3381_3384delAGAA, p.Glu1128Metfs*42; c.1336G > T, p.Glu446Ter; c.291dupC, p.Asn98Glnfs*2; and c.1950C > A, p.Tyr650*) [45,97,98]. As such an important causative gene, understanding the mechanism of action of *STAG3* is crucial. Synapsis mainly occurs during prophase I of meiosis, and it refers to the pairing of homologous chromosomes for DNA exchange of non-sister chromatids, also known as crossover. The synaptonemal complex (SC) acts as a zipper and is responsible for normal synapsis. Cohesin is another essential protein that contributes to establishing SC, ensuring the correct separation of pairing chromatids, DNA repair, and transcriptional regulation [59,99]. Moreover, *STAG3* encodes the corresponding protein, which functions as a subunit of the cohesin complex [44]. Therefore, defects in *STAG3* will lead to abnormal folliculogenesis and, eventually, POI.

#### 4.1.4. Minichromosome Maintenance 8 (MCM8) and Minichromosome Maintenance 9 (MCM9)

MCM8 and MCM9 are homologous to the MCM 2-7 complex, and all belong to the MCM family. Similar to other members of the MCM family, MCM8 and MCM9 contain the highly conserved helicase domain that can open up DNA strands [100]. MCM8 dimerizes with MCM9, giving rise to a hexameric helicase that is responsible for homologous recombination (HR) initiated by DNA double strand breaks (DSB) and facilitating DNA synthesis in a RAD51-dependent manner [46]. The presence of DSB can lead to a loss of DNA integrity, eventually causing follicular apoptosis and degeneration. Therefore, mutations in *MCM8* and *MCM9* are associated with infertility. In female mice, loss of *MCM8* and *MCM9* contributes to sterility [101]. By performing the NGS approach (mainly WES) in POI consanguineous families from various ethnicities (Han Chinese, Arab, Turkish, and Tunisian), new deleterious homozygous mutations in *MCM8* or *MCM9* were identified (c.351_354delAAAG, p.Lys118Glufs*5; c.1483G > T, p.E495*, and c. 482A > C, p.His161Pro) [48,102,103]. These variants cause chromosomal instability and reduce the DNA repairing capacity. In addition, the members of these consanguineous families with heterozygous variants of *MCM8* and *MCM9* are healthy. However, a considerable body of work investigating POI causal genes in several cohorts from different ethnic groups also provides evidence for heterozygous mutations in *MCM8* and *MCM9*, such as c.2488G > A, p.A830T; c.482A.G, p.His161Arg; c.548A.G, p.Asn183Ser; c.686T.G, p.Val229Gly, etc. [41,104,105,106,107,108]. However, not all of the aforementioned mutations have been shown to be harmful. The nature of certain variations has not been determined, and some of the variants are benign. Moreover, The T allele and C allele in two *MCM8* single-nucleotide polymorphisms (SNPs) (rs16991615 and rs451417) were found to be associated with susceptibility to POI in a cross-sectional study [109]. Moreover, Wang et al. reported the first family presenting with a disease-causing *MCM8* mutation (c.724T > C, p.C242R, and c.1334C > A, p.S445*) in adolescence and childhood [47].

#### 4.1.5. DNA Meiotic Recombinase 1 (DMC1)

*DMC1* is an autosomal POI candidate gene that is located at 22q13.1. Its protein product works with RAD51 and RPA to repair DSBs during mammalian meiosis. In this repairing process, DMC1 plays an important role in strand exchange [49]. In addition, DMC1 is associated with fertility capacity. Failure of folliculogenesis and spermatogenesis can be observed in *DMC1* knockout murine models [110]. Moreover, previous studies emphasized its disease-causing role in POI. He et al. investigated *DMC1* mutation in a consanguineous family with POI and non-obstructive azoospermia (NOA) members through WES and found a new missense mutation in *DMC1*, which was the causal mutation in both POI and NOA (c.106G > A, p.Asp36Asn) [50]. Another study performed NGS in 269 POI patients from Caucasian, sub-Saharan African, North African, and Asian origin to screen mutant genes. It was reported that 7% of POI patients in this cohort harbored *DMC1* variants (c.449G > A, p.Gly150Asp; c.598A > G, p.Met200Val) [111]. According to the results of the mutation taster algorithm, c.449G > A is a disease-causing alteration and c.598A > G is a polymorphism. These scientists also used sorting intolerant from the tolerant (SIFT) and PolyPhen-2 algorithms, with all the results indicating that the two DMC variants were damaging, except for the PolyPhen-2 result for c.598A > G (benign). In contrast to these results, many articles report no connection between *DMC1* and POI or female mice fertility. Thus, the level of influence *DMC1* has in female sterility remains unclear [110,112,113].

#### 4.1.6. Myeloid Cell Leukemia-1 (MCL-1)

*MCL-1* exists at chromosome 1 and can encode an antiapoptotic protein, a BCL-2 family member, necessary for regulating the cell cycle. In oocytes, *MCL-1* mutations affect the transition from mitosis to meiosis, resulting in reduced original follicles [51]. Previous research demonstrated the association between female fertility and *MCL-1*. In a study investigating the therapeutic effect of optimized platelet-rich plasma in POI mice, the *MCL-1* expression level was lower in the POI group than in the controls, while *MCL-1* had a higher level of expression after treatment [114]. In contrast to the protective factor, a deleterious factor for female fertility, known as cadmium (Cd), could cause decreased *MCL-1* expression [115]. Cadmium is a heavy metal and is a known toxin with effects on the reproductive system. It has many effects on various cellular processes. In addition, it was demonstrated in other studies that *MCL-1* is expressed in primordial and preantral follicles, contributing to the normal development of follicles [116]. Moreover, MCL-1-depleted mice exhibited similar presentations to POI patients [52]. However, the authors in one work of literature failed to reveal the causal connection between the *MCL-1* gene and Chinese idiopathic POI patients [51].

#### 4.1.7. MutS Homolog 4 (MSH4) and Muts Homolog 5 (MSH5)

Recently, *MSH4* (1p31) and *MSH5* (6p21.31) became POI candidate genes, due to their roles in chromosomal synapsis and meiotic recombination through forming a dimer [38]. Both are members of the MutS family, which is associated with DNA mismatch repair. By performing WES in Iranian, Chinese, and Colombian families, two homozygous missense variants from *MSH4* and *MSH5* (NM_002440.4: c.2261C > T, p.Ser754Leu; ENST00000375755: c.1459G > T, p.D487Y) and a homozygous donor splice-site *MSH4* variant (p.Ile743_Lys785del) were identified [53,117,118]. These two mutations in *MSH4* are harmful to the ATP binding site, affecting the gene’s normal function. In a *MSH5* variant-finding study, the scientists also identified infertile female mice with a mutation of the gene homologous to *MSH5* p.D487Y. In addition, three other unconfirmed heterozygous *MSH5* variants were also revealed by screening 200 patients with sporadic POI. Moreover, two novel mutations in *MSH5* were recently identified in two POI cohorts (c.1264C > T, p.Arg422Cys and c.C1051G, p.R351G) [54,119]. Scientists in these studies explored the effects of these variants via yeast assay and C. elegans, respectively. However, only the c. C1051G; p.R351G variant, tested using C. elegans, exhibited obvious defects.

#### 4.1.8. Meiosis Specific with OB Domain (MEIOB)

*MEIOB* (16p13.3) is another mandatory gene associated with RAD51 and DMC1 stabilization, meiotic homologous recombination, and DSB repairing. The MEIOB protein binds to single-strand DNA and dimerizes with spermatogenesis-associated 22 (SPATA22) [55]. Both are important for fertility in males and females. Two recent studies identified two homozygous *MEIOB* mutations that affect female fertility in Arab and Pakistani consanguineous families (c.1218G > A and c.683-1G > A) [120,121]. They also revealed infertile female mice with homozygous deletion of *MEIOB*. The mutation identified in the Arab family was responsible for the failure of MEIOB–SPATA22 binding and then POI, while the other variant was unrelated to the interaction between these two proteins. As regards the relationship between MEIOB and SPATA22, another study showed that SPATA22 contributed to the localization of MEIOB [122]. In addition, a research group found many new pathogenic homozygous *MEIOB* variants (c.258_259del, c.1072_1073del and c.814C > T) in three consanguineous Chinese families containing POI and NOA patients [56]. All the variants give rise to truncated proteins, the functions of which are affected.

#### 4.1.9. PSMC3 Interaction Protein (PSMC3IP)

*PSMC3IP* (17q21.2) is also known as homologous-pairing protein 2 (HOP2), and its protein product forms a complex with Mnd1 to activate DMC1 and Rad51 recombinases using the same protein domains [57]. Therefore, it is also necessary in HR and fertility. In *PSMC3IP* knockout female mice, smaller ovaries and a depletion of follicles can be observed [40]. In female members with POI and primary amenorrhea from a Yemeni consanguineous family, a deleterious homozygous stop gain mutation of *PSMC3IP* (c.489 C.G, p.Tyr163Ter) was identified, which caused the partial deletion of the C-terminal portion in the PSMC3IP protein [58]. This deletion led to failed interaction with RAD51 and DMC1, contributing to impaired HR and DNA repair. Aside from the POI patients with primary amenorrhea, two compound heterozygous *PSMC3IP* mutations (c.206_208delAGA and c.189 G > T) in a secondary amenorrhea POI patient was discovered in 2022 [123]. Another two compound heterozygous mutations (c.597 + 1G > T and c.268G > C p.D90H) in a Chinese POI woman have also been reported [124]. Conversely, no *PSMC3IP* mutations were observed in a POI cohort from Sweden, which indicates that more studies are required to establish causal *PSMC3IP* variants in POI cohorts from different populations and containing different ethnicities [125].

#### 4.1.10. Fanconi Anemia Complementation (FANC) Group Genes

Fanconi anemia (FA) is a heterogeneous recessive human genetic disease caused by mutations of one of the genes in the FANC group. These mutations usually lead to impaired meiosis and folliculogenesis [59]. Among genes in the FANC group, many are associated with female fertility. It was reported that *FANCI*, *FANCB*, *FANCA*, and *FANCE* mutations could affect fertility in female mice [126,127,128,129]. In addition, the *FANCL*, *FANCA*, and *FANCM* variants were identified in non-syndromic POI patients (c.1048_1051delGTCT, p.Gln350Valfs*18; c.739dupA, p.Met247Asnfs*4; c.1772G > A, p.R591Q; c.3887A > G, p.E1296G; and c.5101C > T, p.Gln1701*) [60,130,131,132].

#### 4.1.11. Other Genes

There exist other rare and novel POI candidate meiotic genes (*SYCP2L*, *HSF2BP*, and *ZSWIM7*) [61,62,63,64,133]. These studies expand the range of POI-causing genes.

### 4.2. Transcription Factor

Although transcription factors cannot directly participate in many vital biological processes, they all play crucial parts in regulating physical activities by controlling various target genes. As regards female fertility, they control the expression time, the site, and the expression level of reproductive genes to ensure the smooth functioning of every reproductive process.

#### 4.2.1. Nuclear Receptor Subfamily 5, Group A, Member 1 (NR5A1) or Steroidogenic Factor-1 (SF-1)

*NR5A1* is located at 9q33.3 and encodes an orphan nuclear receptor. It is expressed in the gonads and adrenals, and the NR5A1 protein acts as a transcription factor, regulating the expression of genes involved in steroidogenesis, reproduction, and gonadal and adrenal development in human, such as Eps15 homology domain-containing protein 3 (EHD3), anti-Mullerian hormone (AMH), and Wilms’ tumor 1 (WT1), etc. In addition, a previous research study emphasized the importance of screening *NR5A1* variants for POI women with a family member suffering from disorders related to gonadal development [134]. In women, *NR5A1* is mainly expressed in the granulosa and theca cells to control ovarian folliculogenesis and steroidogenesis [135], but the POI pathogenic role in *NR5A1* variants is controversial. Certain NR5A1 variants (p.Ser54Arg, p.Pro198Leu, p.Pro129Leu, and p.Gly123Ala) found in POI patients are not pathogenic, and the pathogenicity of some mutations, such as c.437G > C, IVS4-20C > T, has not been confirmed via functional tests [136,137]. There also exist other NR5A1 variants that have been shown to be deleterious. For example, a heterozygous missense *NR5A1* variant (c.74A > G, p.Y25C), a rare missense *NR5A1* variant (c.1063G > A, p.(Val355Met)), and the p.Val15Met *NR5A1* variant were demonstrated to be deleterious in POI patients [65,138,139].

#### 4.2.2. Newborn Ovary Homeobox Gene (NOBOX)

*NOBOX* (7q35) is expressed in oocytes and granulosa cells and is responsible for early folliculogenesis. In one study, which established a *NOBOX*-deficient female mice model, a reduced number of primordial follicles (due to abnormal germ cell cysts) and adherens junctions between unseparated oocytes was observed, which indicated the important role of *NOBOX* in female mice fertility [66]. In addition, it also indicated the significance of oocyte–somatic cells signaling for mice POI. *KIT-L* is one of the target genes of NOBOX and is present in granulosa cells. Kit-L can transmit the signals of granulosa cells to oocytes via the phosphatidylinositol 3-kinase/AKT pathway [140]. Moreover, other ovarian-specific genes, including growth and differentiation factor 9 (GDF9), bone morphogenetic protein 15 (BMP15), and POU class 5 Homeobox1 (POU5F1) are under the control of NOBOX [141,142,143]. Aside from regulating target genes, NOBOX can also interact with FOXL2; however, mutant *NOBOX* affects this interaction, leading to POI [144]. A previous study conducted in China demonstrated that a homozygous *NOBOX* mutation (c.567delG, p.T190Hfs*13) failed to arrest G2/M, causing disrupted cell cycles in meiotic oocytes [67]. According to recent articles, *NOBOX* is a strong candidate gene for causing POI in humans, due to its high prevalence in different POI populations (5.6–6.5%) [140,142,145]. In addition, these researchers confirmed various *NOBOX* variants (p.Gly91Thr, p.Gly111Arg, p.Arg117Trp, p.Lys371Thr, p.Pro619Leu, p.Gly91Trp, p.R44L, p.G91W, p.G111R, p.G152R, p.K273*, p.R449*, and p.D452N) to be deleterious.

#### 4.2.3. Spermatogenesis-and Oogenesis-Specific Basic Helix-Loop-Helix 1 (SOHLH1) and 2 (SOHLH2)

*SOHLH1* (9q34.3) and *SOHLH2* (13q13.3) are only expressed in germ cells, primordial follicles, and primary follicles, and their protein products can form a heterodimer, which functions as master–master regulators of other master transcription factors to ensure the normal development of gonadal glands and primordial follicles [68,70]. In a murine model, deficiency of at least one of these two genes caused reduced ovarian sizes and a decreased number of primordial follicles and primary follicles [146]. *SOHLH1* and *SOHLH2* are POI candidate genes, but few studies have investigated their POI-causative roles in the last 10 years. POI-related *SOHLH1* variants (p.Ser317Phe, p.Glu376Lys, and c.*118C > T) were first observed in 2015 [69]. These three deleterious variants were identified in Chinese POI cases, while the variants in Serbian POI cases were synonymous. Two causative variants, p.Ser317Phe and p.Glu376Lys, altered the transactivation capacity of the *SOHLH1* gene, resulting in lower expression levels of its target genes, including LIM homeobox 8 (LHX8) and zona pellucida glycoprotein 1 (ZP1) and 3 (ZP3). Additionally, scientists identified five heterozygous nonsynonymous *SOHLH2* mutations in a POI cohort of the same ethnicity as the cohort discussed before [70]. They were c.235G > A, p.Glu79Lys; c.314A > G, p.Glu105Gly; c.961A > C, p.Thr321Pro; c.360A > T, p.Leu120Phe and c.610C > T, p.Leu204Phe. Among these variants, only c.235G > A and C.314A > G proved to be deleterious. Moreover, this study firstly demonstrated the association between *SOHLH2* and idiopathic POI. Furthermore, two more recent studies performed next-generation sequencing in a POI cohort containing 100 patients and a cohort containing 36 Turkish families with POI members, and both showed deleterious *SOHLH1* mutations [72,147].

#### 4.2.4. LIM Homeobox 8 (LHX8)

*LHX8* (1p31.1) is expressed in germ cells and is essential for early oogenesis. In mice ovaries without the *LHX8* gene, remarkably decreased expression levels of other genes associated with oogenesis are observed, which indicates that LHX8 functions as a transcription factor [71]. This study also demonstrates that LHX8, SOHLH1, and FIGLA can interact with each other, forming a nuclear complex and regulating the expression of various oogenesis-related genes. Aside from mice, LHX8 contributes to oogenesis in rainbow trout [148]. In 2016, Bouily reported a missense mutation of *LHX8* (c.974C > T, p.A325V) in a Caucasian POI women cohort and verified its damaging effect on POI [72]. In addition, a recent study also revealed a disease-causing *LHX8* variant (c.974C > T p.Ala325Val) in POI patients [111]. However, it was reported that there were no causative *LHX8* mutations in 95 US Caucasian women with POI [149]. Therefore, *LHX8* may be a relatively rare candidate POI-causing gene in the US Caucasian population, and more research is needed to this end.

#### 4.2.5. Folliculogenesis Specific BHLH Transcription Factor (FIGLA)

*FIGLA* is located at 2p13.3. It is mainly expressed in female germ cells and plays an important role in oogenesis. In female mice without *FIGLA*, the genes involved in different processes (meiosis, growth, and differentiation) of oogenesis are downregulated, such as *Rad51*, *SYCP3*, *NOBOX*, *LHX8*, *SOHLH1*, and *SOHLH2*, which indicates an association between *FIGLA* and female fertility [71]. In addition, zona pellucida glycoprotein genes, which are important in folliculogenesis, also fall under the control of FIGLA [14,28]. Aside from oogenesis disruption, impaired secondary follicles maturation may also be a pathogenesis of POI caused by *FIGLA* variations. A recent article showed that FIGLA had the ability to support the development of secondary follicles in mature female mice [150]. Moreover, many researchers have attempted to evaluate the impact of *FIGLA* mutations on POI, with various deleterious *FIGLA* variants (c.364del p.Glu122Lysis*45 and c.2 T > C, p.Met1Thr) in consanguineous and non-consanguineous families being reported [73,139,151]. All these variants were homozygous with the autosomal recessive inheritance mode.

### 4.3. Signaling Molecules and Receptors

Signaling molecules, such as hormones, have the ability to transmit information between cells to regulate cellular activities through binding to receptors outside or inside cells. A variety of hormones and receptors are involved in many important processes related to female reproduction and fertility capacity. Their expression levels in human bodies can be considered indicators of female reproductive status and have strong diagnostic significance.

#### 4.3.1. Follicle-Stimulating Hormone Receptor

Follicle-stimulating hormone (FSH) is a glycoprotein essential for the development of antral follicles. It also promotes hormone secretion, such as estradiol, and indicates the ovarian reserve level by binding to FSHR. Therefore, mutations in the *FSHR* gene may negatively affect FSH-FSHR signaling, arresting folliculogenesis and causing POI. Data from several studies identified detrimental *FSHR* variants associated with POI. A novel missense *FSHR* mutation (I423T), a homozygous *FSHR* variant (c.1253T > G, p.Ile418Ser), and two compound heterozygous missense *FSHR* variants (c.646 G > A, p.Gly216Arg and c.1313C > T, p.Thr438Ile) were observed in different POI women from various ethnicities [74,75,152]. All these mutations can affect the expression of *FSHR*, forming different partial FSHRs at granulosa cell membranes with impaired function. However, various studies failed to establish a causative relationship between *FSHR* mutations and some POI cases [153,154,155]. Woad et al. found no harmful *FSHR* variants in exons 7 and 10 from a population of POI women from New Zealand. Another two studies revealed both positive and negative results. In 192 Han Chinese participants with POI, the p.L5971 variant was deleterious, while mutation p.M265V was harmless. In addition, according to a meta-analysis, a *FSHR* polymorphism (rs6166) was regarded as a genetic biomarker exclusively in POI cases from Asia. In summary, *FSHR* is a rare POI candidate gene in certain ethnic populations, and researchers should focus on screening *FSHR* mutations in more POI patients from different countries to provide further evidence of the role of *FSHR* in POI.

#### 4.3.2. Inhibin Family

*Inhibin* genes can be divided into two types: inhibin alpha (INHA) (2p35) and inhibin beta (INHB), which includes inhibin beta A (INHBA) (7p15-p13) and inhibin beta B (INHBB) (2cen-q13). They encode the corresponding subunits to give rise to two forms of dimeric glycoproteins (inhibin A and inhibin B) belonging to the transforming growth factor-beta (TGF-beta) family. As regards the mechanism of function, inhibins cause downregulated FSH through the inhibin/beta glycan complex, and they compete with activins (a FSH promoting factor) for binding to ACVR2 [33]. Inhibins and activins act opposingly, but regulate the FSH level together, associating with folliculogenesis. However, there is only evidence of a causative link between *inhibin* genes and POI. In a 136 Korean POI population, the effect of the T–G haplotype and the CT + TT/GG genotype of two *INHA* polymorphisms (c.-16C > T and c.-124A > G) were related to the susceptibility to POI [156]. In a recent study, both homozygous and heterozygous c.769G > A variants of *INHA* were shown to be correlated with POI patients from a Kashmiri cohort [76]. This study also showed increased FSH levels in POI patients, which caused accelerated follicle recruitment and premature loss of the follicular pool. This phenomenon is consistent with the characteristics observed in female mice, with an inactivated *INHA* mutation. An increased FSH expression level, a high ovulation rate, and impaired ovarian function appeared in the affected mice [157]. However, *INHA* is heterogeneous in different populations. For example, an *INHA* variant (G769A) is uncommon in Brazilian and Argentina POI patients, while there is a higher prevalence of G796A in Indian and Italian POI women [158]. Moreover, Ma et al. suggested that *INHBB* mutation (c.1095C > A) may be the cause of POI cases in Chinese Hui women [159]. Further research is warranted to elucidate the causative role of *INHBB*.

#### 4.3.3. Anti-Mullerian Hormone (AMH) and Anti-Mullerian Hormone Receptor 2 (AMHR2)

The *AMH* gene is located at chromosome 7, and its protein product can only bind to AMHR2. This ensures the successful control of the AMH signaling pathway, which plays a crucial role in the development of follicles. It has been suggested that AMH can work synergistically with INHA to prevent the formation of FSH-initiated progesterone and estradiol and eventually contribute to mice reproduction [77]. Various recent studies investigated the connection between *AMHR2* and POI. In 2014, two novel missense *AMHR2* variants (p.I209N and p.L354F) were identified in a Chinese POI cohort [78]. However, the functional assays were not performed to establish their role in AMH signaling. Their effects were confirmed by silico analysis in 2016 [160]. The scientists chose another Chinese Han POI patient population and found a novel missense *AMHR2* variant, p.Ala17Glu (A17E). Moreover, they demonstrated that p.I209N, p.L354F, and p.Ala17Glu (A17E) were all detrimental, but only the 1209N mutation harmed the AMH signaling pathway. In addition, it has been shown that other deleterious rare missense *AMH* mutations are associated with POI [161].

#### 4.3.4. Bone Morphogenetic Protein 15 (BMP15) and Growth Differentiation Factor 9 (GDF9)

*BMP15* (Xq11.22) and *GDF9* (5q31.1) belong to the TGF-beta superfamily and are expressed exclusively in oocytes. They are essential for many processes associated with female fertility via forming heterodimers or homodimers to induce downstream signaling pathways [162]. BMP15 is involved in promoting follicle development during primordial gonadotropin-independent phases, controlling the sensitivity of GCs to FSH and ovulation, and avoiding depletion of GCs. GDF19 is responsible for the maturation of follicles from the primary to the secondary stage, stimulating GC proliferation, regulating various GC enzymes related to cumulus expansion, etc. [1,80]. To date, many researchers have attempted to investigate the impact of *GDF9* mutations on POI. Two homozygous variants from *GDF9* (c.783delC and c.604C > T, p.Gln202*) were discovered in one Brazilian POI patient and two Caucasian siblings with POI [81,163]. These two mutations caused truncated GDF9 proteins, which negatively affected the biological function of the GDF9 protein. Furthermore, the first likely POI-causing mutation influencing the regulatory region of *GDF9* was identified in 2014 via high-resolution array comparative genomic hybridization (CGH) analysis [164]. As regards *BMP15*, its heterozygous variants are the second-most frequent causative mutant gene causes of POI. Numerous studies have assessed the pathogenesis of *BMP15* heterozygous variants. Similar to *GDF9*, *BMP15* mutations were found in two POI siblings (c.791G > A, p.R264Q, and c.1076C > T, p.P359L) [79]. These variants conformationally affect protein-surrounding water molecules and the thermal stability of BMP15. Scientists also conducted in vitro cell line experiments, which showed impaired BMP15 function. Another two studies also identified deleterious heterozygous *BMP15* variants (p.N103K, p.M184T, and c.406G > C (V136L) in POI patients [165,166]. Moreover, homozygous *BMP15* mutations are also implicated in POI. Zhang et al. firstly predicted the pathogenic role of its homozygous variants (c.G1070A, p.C357Y) in a Chinses POI girl from a consanguineous family [167]. More recently, another biallelic missense variant (c.1076C >T, p.Pro359Leu) was identified in a case report [168]. Although the prevalence of *GDF9* and *BMP15* variants in POI patients is high, there still exist idiopathic POI cases without causative mutations in these two genes, indicating the presence of heterogeneity [169].

#### 4.3.5. Other Novel Candidate Genes

Many studies have focused on finding new POI-causing genes, which provide idiopathic POI patients with better diagnoses and treatments and broaden the spectrum of the genetic factors implicated in POI. For example, linkers for the activation of T cells (LAT), vascular endothelial growth factor A (VEGFA), and bone morphogenic protein receptors 1A (BMPR1A) and 1B (BMPR1B) were first linked to POI in the 5 five years [82,83,84].

### 4.4. RNA Metabolism and Translation

RNAs connect DNAs and proteins. Therefore, impaired RNA activities, including metabolism and translation, have an adverse impact on protein production. These proteins may play important roles in various biological processes, such as folliculogenesis, steroidogenesis, cell proliferation, apoptosis, etc.

#### 4.4.1. Nanos C2HC-Type Zinc Finger 3 (NANOS3)

It is known that the migration, development, and maintenance of primordial germ cells (PGCs) cannot be finished successfully without the involvement of *NANOS3* (19p13.13). *NANOS3* encodes an RNA-binding protein with anti-apoptotic and translation repressive abilities. In addition, mutations in *NANOS3* are associated with POI, based on the function of this gene. According to a recent article, a homozygous mutation of *NANOS3* (c.358G > A, p.Glu120Lys) was found in two sisters from a Brazilian POI cohort using a mutational analysis [85]. This mutation is located at the zinc finger domain and affects the interaction of the NANOS3 protein and its target mRNA. Moreover, in vitro experiments were also conducted to evaluate and confirm the accelerated apoptosis of PGCs caused by a p.Glu120Lys mutation. However, no causative mutation of *NANOS3* was found in another population of Brazilian women with POI, which indicates the heterogenicity of *NANOS3*, depending on different populations [170]. Aside from the Brazilian population, a novel pathogenic heterozygous *NANOS3* variant (c.457C4T; p.Arg153Trp) was identified in a Chinses POI cohort [171]. The researchers also established heterozygous and homozygous p.Arg153Trp knockout mice models and identified the relationship between the dosage of functional *NANOS3* and the normal development of PGCs.

#### 4.4.2. Eukaryotic Translation Initiation Factor 4E Nuclear Import Factor 1 (EIF4ENIF1)

*EIF4ENIF1* (22q11.2) encodes a nucleocytoplasmic shuttle protein, which can transport a translation-induced protein called eIF4E to repress translation via interrupting eIF4E–eIF4G binding [86]. The disrupted translation activities caused by *EIF4NIF1* mutations lead to increased mRNA expression and stability, which may result in disrupted ovarian follicles and enhanced oocyte apoptosis [87]. It has been demonstrated that *EIF4ENIF1* mutations are implicated in POI. The first heterozygous *EIF4NIF1* mutation (c. 1286C > G, p.Ser429X) was found in all POI patients from the same family in 2013 [87]. An additional pathogenic heterozygous variant (c.2525A > C, p.Q842P) was identified in both diminished ovarian reserve (DOR) and POI cases through whole-exome and Sanger sequencing [8]. Moreover, the secondary structure of the EIF4ENIF1 protein with a p.Q842P mutation revealed the abnormal structure and length of the alpha-helix, which can influence EIF4ENIF1’s ability to regulate the translation of mRNA. In addition, two more rare *EIF4ENIF1* variants (c.9_11delGAG, p.R4del and c.2861G > C p.G954A) from two Han Chinese POI women were reported in 2022 [172]. However, multiple bioinformatic tools showed that only p.G954A was detrimental.

## 5. Single Gene Mutations in Syndromic and Pleiotropic POI

Unlike non-syndromic POI, POI can also be present in syndromic or pleiotropic Mendelian diseases. Therefore, the gene mutations that cause POI-relevant Mendelian inheritable disorders can be considered to be the genetic factors contributing to POI.

### 5.1. Fragile X Mental Retardation 1 (FMR1)

The fragile X mental retardation protein (FMRP) encoded by *FMR1* (Xq27.3) acts as a type of RNA-binding protein and plays an important role in regulating translation. Any deleterious mutations in *FMR1* can lead to abnormally expressed FMRP. However, the expansion of trinucleotide (CGG) repeats at 5′UTR in *FMR1* is the most common cause [173]. This expansion can be regarded as a pathogenic variant that contributes to different and unrelated pathologies, depending on the expanded length of CGG repeats (pleiotropy). Normally, there is an AGG triplet after every nine to ten CGG repeats, which is called AGG interruption. This is due to the existence of a sufficient number of AGG triplets for the number of CGG repeats to be maintained within a stable range, i.e., the normal alleles contain 4–55 CGG repeats [174]. The most frequent CGG length varies among different populations [175,176]. Women with 45–54 CGG repeats and 55–200 CGG repeats have gray zone (GM)/intermediate alleles and premutation (PM) alleles, respectively. Both are related to fragile X-associated primary ovarian insufficiency (FXPOI) in women from various countries, such as Turkey, India, Argentina, Brazil, and China [175,176,177,178,179,180]. According to these studies, the prevalence of POI patients with GM and/or PM alleles ranges from 1% to 9.6%. This mutation is rare in Chinese and south Indian POI patients. Nevertheless, the premutation range of the *FMR1* gene is the most common aetiology among POI cases caused by single gene mutations [7,14]. Approximately 16% of women with PM alleles suffer from POI [181]. In addition, a higher percentage (34.2%) was reported in a large Turkish cohort [177]. As compared to PM alleles, the GM/intermediate repeat size seems less implicated in POI. One recent meta-analysis failed to establish a connection between the GM CGG repeat length of *FMR1* and susceptibility to, or severity of, POI [182]. Moreover, several studies also indicated that there was no correlation between GM alleles and POI [183,184]. In addition, the fully mutated alleles of *FMR1* (>200 CGG repeats) initiated via maternal transmission from premutation or an intermediate size are significantly correlated with fragile X syndrome [173].

To date, numerous researchers have investigated how *FMR1* variants cause FXPOI, and they have attempted to explain the potential mechanisms involved. Recently, a case report reported an FXPOI woman who became spontaneously pregnant with two healthy babies [185]. In this study, a *FMR1* premutation mice model was also established. These experimental mice carried a normal number of primordial follicles, a reduced number of antral follicles and corpora lutea, and an increased number of atretic large antral follicles. Therefore, it was the disrupted follicular function, not the exhausted primordial follicles, that led to FXPOI. Furthermore, another murine model in which premutation *FMR1* alleles were introduced was established by scientists in 2012 [186]. It showed the impaired development of immature follicles, except for primordial follicles, which is consistent with the results of the aforementioned article. In addition, impaired luteinizing hormone (LH) and Akt/mTOR-mediated biological pathways were also associated with inducing FXPOI. Finally, abnormal *FMR1* alleles were shown to produce a higher risk of reduced functional ovarian reserve (FOR) in 30–38-year-old women because they were associated with skewed X-chromosome inactivation, which contributed to a low level of AMH [187].

### 5.2. Forkhead Box L2 (FOXL2)

*FOXL2* is located at 3q23, and it can be found in the ovary. It encodes the forkhead transcription factor, which is essential for maintaining ovarian somatic cells’ (GCs and theca cells) identities by preventing the transdifferention to their testicular counterpart and regulating the expression of genes involved in estrogen production, folliculogenesis, and steroidogenesis [188,189]. Mice without *FOXL2* manifest impaired maturation of follicles and altered gonadotrophic production, thus affecting their fertility [190,191]. In addition, the high expression level of *FOXL2* in female mice can also affect their reproductive ability by destroying the differentiation of granulosa and theca cells, influencing steroidogenesis, etc. [192]. *FOXL2* mutations are associated with a type of rare and autosomal dominant syndrome, known as blepharophimosis-ptosis-epicanthus-inversus syndrome (BPES). Two types of BRES are classified according to the consequences of different mutations. POI can only be identified in BPES type I [188]. Therefore, FOXL2 is responsible for syndromic POI, and it represents the first autosomal gene related to syndromic POI [191]. A great deal of previous research into *FOXL2*-implicated syndromic POI is focused on Chinese populations. In Chinese families, scientists discovered various disease-causing *FOXL2* variants (c.307C > T p.Arg103Cys, c.462_468del, and c.988_989insG) from BPES type I members with POI [193,194]. Moreover, researches showed two new *FOXL2* mutations in Chinese families with BPES type II, but failed to link this to POI [195]. However, a case report conducted in 2019 revealed one of the two aforementioned *FOXL2* mutations, showing that the *FOXL2* variant (c.223C > T p.Leu75Phe) was associated with women with typical POI hormonal alteration from a BPES type I Polish family [196]. Aside from syndromic POI, *FOXL2* mutations are also responsible for non-syndromic POI [72,197].

### 5.3. Galactose 1-Phosphate Uridyl Transferase (GALT)

The *GALT* gene exists at 9q13 and encodes one of the enzymes necessary in the main galactose metabolism pathway (the Leloir pathway). The other two enzymes are galactokinase (GALK) and UDP galactose 4-epimerase (GALE). The functional-affecting disorders of any of these three enzymes can lead to impaired galactose metabolism and eventually galactosemia. Recently, pathogenic mutations in the galactose mutarotase (GALM) gene were identified in patients with unexplained galactosemia, giving rise to a new type of galactosemic [198]. Classical galactosemia (GC) and Duarte galactosemia (DG) are caused by mutations in *GALT*. As a symptom of hypergonadotropic hypogonadism, POI is one of the long-term complications of GC, while it has not been observed in DG female patients [199,200,201]. Moreover, the prevalence of POI in GC patients is high. Over 90% of GC patients exhibited signs of POI [202]. Thus far, the exact mechanism of POI in galactosemia has not been discovered. However, studies over the past 10 years have highlighted many potential mechanisms, including (1) ovarian damage caused by the toxic effects from accumulated galactoses and/or their metabolisms; (2) alteration of the function of FSH resulting from impaired glycosylation; (3) defects in the development of follicles, germ cells functions, and steroidogenesis, due to low levels of UDP-glucose (UDP-Glc) pyrophosphorylase and UDP galactose (UDP-Gal); (4) aberration of cell signaling pathways, such as the PI3K/AKT/mTOR signaling pathway; and (5) epigenetic mechanisms [203,204]. As regards specific *GALT* variants in galactosemia patients with POI, p.Q188R and p.K285N are the most common mutations, accounting for 70% of cases [7]. Although a low pregnancy rate (5–10%) is generally reported among POI patients, women with POI and galactosemia can still become pregnant spontaneously [205,206].

### 5.4. Autoimmune Regulator (AIRE)

Polyendocrinopathy candidiasis ectodermal dystrophy (APECED)/autoimmune polyendocrinopathy syndrome type I (APS1) is a rare monogenic autoimmune disease. It is characterized by two out of three of the following clinical manifestations: chronic mucocutaneous candidiasis (CMC), hypoparathyroidism, and primary adrenal insufficiency (Addison’s disease). In addition, deleterious mutations in the autoimmune regulator gene (AIRE) can confirm the diagnosis of APECED [207]. Aside from these three major clinical manifestations, other features, such as POI, are also observed in APECED patients. The literature review revealed that, in a large cohort of APECED patients from various countries, the prevalence of gonadal failure ranged from 0 to 70% [208]. It was also reported that the female-to-male ratio of hypergonadotropic hypogonadism was 7:1 in North America. Moreover, there is a considerable body of literature on finding *AIRE* variants in APECED patients. Various homozygous and heterozygous pathogenic *AIRE* variants (c.967_979delCTGTCCCCTCCGC, p.(L323SfsX51); c.995 + (3_5)delGAGinsTAT, NM_000383.2: c.623G > T, NP_000374.1: p.Gly208Val; c.967_979del13bp; c.396G > C (p.Arg132Ser; p.R132S) and (c.47C > T, p.Thr16Met)) were observed in APS1 women with POI as a symptom [209,210,211,212,213]. As regards the toxic effects of *AIRE* mutations on the ovaries, loss-of-function *AIRE* can cause ovarian autoimmune disease by repressing the antigen expression levels specific to the ovaries. In female mice without *AIRE*, follicular depletion and an exhausted ovarian reserve were observed, indicating a potential mechanism in female infertility [214].

### 5.5. Other Syndromic Disorders

Aside from the pleiotropic and syndromic disorders that were discussed in detail previously, there are also many studies from the last 10 years that revealed gene mutations that cause various syndromes (ataxia telangiectasia, Nijmegen breakage syndrome, Alagille syndrome, Mulibrey nanism disorder, and congenital disorders of glycosylation) that are characterized with POI [108,215,216,217,218].

## 6. Mitochondrial Dysfunction in POI

The mitochondrion is a common organelle in most eukaryotic cells. It plays an essential role in producing energy via oxidative phosphorylation. Therefore, abnormalities in mitochondrial functions are associated with a wide range of human diseases, including POI. In fact, any perturbations in mitochondria can severely affect the ovaries because the ovaries contain the maximum number of mitochondria. As regards the genetic factors related to mitochondrial dysfunction, mutations in nuclear and mitochondrial genes are responsible for POI. According to recent studies, required for meiotic nuclear division 1 homolog (RMND1), mitochondrial cytochrome c oxidase 1 (MT-CO1), nuclear-encoded gene mitochondrial ribosomal protein S22, (MRPS22), caseinolytic peptidase B (CLPB), and mitochondrial transcription factor A (TFAM), variants were reported in POI, and their pathogenic roles were confirmed [219,220,221,222,223] (Table 2). In addition, mutations in the genes that regulate mitochondrial activities or functions can also lead to POI. In 2021, Feng et al. demonstrated that a leucyl-tRNA synthetase 2 (LARS2) variation increased the amount of mitochondrial reactive oxygen species (ROS), decreased the number of mitochondrial DNA (mtDNA) copies and ATP, and reduced the expression level of a mitochondrial fusion-related gene known as mitofusin-2 (Mfn-2), which is one of the pathogenic mechanisms of POI (impaired mitochondrial function in granulosa cells) [224]. Furthermore, by establishing a POI mice model, scientists observed conformational changes (swollen mitochondria, decreased matrix density, and marginally shifted cristae) in mitochondria from granulosa cells of POI mice [225]. In addition, a low level of mitochondrial oxidative phosphorylation complexes (OXPHOS) also existed in the POI group. This resulted from a Sirtuin 1 (SIRT1) mutation. Additionally, a new variation in alanyl-tRNA synthetase 2 (AARS2) was identified in a Chinese consanguineous family, resulting in impaired translation activity in mitochondria implicated in POI [226].

## 7. Non-Coding RNA in POI

Establishing the role of non-coding RNAs (ncRNAs) in POI is an emerging research area. It will help us in understanding the genetic effects of POI and provide important information on POI pathogenesis. NcRNAs can be divided into small ncRNAs (sncRNAs) and long ncRNAs (lncRNA), and they contribute to regulating various biological processes, such as cell proliferation and apoptosis, rather than giving rise to proteins [227]. They usually have an influence on ovarian function by interacting with their corresponding target genes. MicroRNAs (miRNAs), as one essential sncRNA type, are closely related to POI. Increasing levels of both microRNA-379-5p and microRNA-127-5p were observed in the GCs of biological POI (bPOI) patients. They had an adverse impact on GCs proliferation and the ability to repair DNA damage through targeting poly ADP-ribose polymerase 1 (PARP1) (microRNA-379-5p), X-ray repair cross complementing 6 (XRCC6) (microRNA-379-5p), and high mobility group box 2 (HMGB2) (microRNA-127-5p) [228,229]. In addition, the suppression of the B-cell lymphoma 9 (BCL9) level initiated by upregulated microRNA-122-5p was shown to promote GCs apoptosis in POI mice [230]. Distinct from the miRNA mentioned previously, microRNA-22-3p acts as a protective factor for POI, which was demonstrated in a receiver operating characteristic (ROC) curve, logistic binary regression, and bioinformatics analysis [231]. Aside from miRNAs, lncRNAs are also critical in POI development. According to a recent study, accelerated GCs apoptosis was observed due to downregulated lncRNA *HCP5* in a Chinese bPOI cohort, which led to a lower nuclear level of *YB1* and eventually induced the obstructed transcription of a critical POI candidate gene, known as *MSH5* [232]. Moreover, the lower expression of another lncRNA, known as *PVT1*, was also shown to disrupt ovarian function in POI patients. These deleterious effects were demonstrated in a POI mice model [233].

## 8. Discussion and Conclusions

Many strategies are used to establish the genetic factors related to POI, and these factors will guide future POI prediction, diagnosis, and treatment protocols. Chromosome analysis (karyotyping) is the main method for discovering abnormalities in chromosomes. Approximately 10–13% of POI cases are associated with chromosomal abnormalities, such as mosaicism 45, X/46, XX. Therefore, chromosome analysis is not only helpful in identifying POI-related chromosomal variations, but also plays an important role in the clinical evaluation and diagnosis of POI. Moreover, for small duplications and deletions that cannot be detected under a microscope, array comparative genomic hybridization (CGH) is the better choice. In 2022, CGH was performed in a Tunisian family [234]. The study suggested that EIF1AX duplication might lead to POI after a familial tandem duplication in Xp22.12 was identified. In addition to chromosome variation, GWAS, WES, and NGS are used to more effectively and efficiently detect single gene mutations than classical methods. In addition, the strong POI component and the similar genetic background among family members make family analyses important in investigating the genetic causes of POI. As a result of GWAS, many loci that are potentially responsible for POI have been disclosed in Chinese, Korean, and Dutch women [14]. Moreover, via NGS, various POI-related genes involved in meiosis and DNA repair have been identified, enriching the genetic etiology of POI [235]. The contributions of GWAS and NGS are indisputable, and they will certainly be applied more widely in the future. As regards WES, it has revealed many POI-related genes involved in DNA damage repair and HR, and its application will also be of great importance in the future. Recently, several in vitro cellular models have been successfully used to demonstrate that certain rare genomic variants can cause mutations and dysfunction of the corresponding proteins, thus confirming the association between these variants and POI [236,237]. Once new deleterious variations are found, they can be used to predict the age of menopause [236]. Thus, this review may be useful in large genetic screening research for POI and in regulating fertility in women. Taken together, the genetic factors of POI confirmed in different tests can be used as the basis for POI diagnosis and risk prediction protocols. Reproductive and genetic counseling is essential for women with POI or those at risk of POI. It can help women select when to attempt pregnancy and increase the possibility of pregnancy through other methods, such as assisted reproductive technologies, oocyte or embryo cryopreservation, etc. [11].

In order to fully understand the genetic causes of POI, many challenges remain. Future research will be designed to this end. First, the mechanisms involved in POI that are caused by certain genetic factors are unclear. For example, although TS is ubiquitous in POI patients, the exact pathogenesis of TS-causing POI remains unknown. Second, less attention has been paid by scientists in recent years to certain areas, such as TXS and structural chromosomal abnormalities. Third, many genes, such as *MCL-1*, have been shown to be associated with POI, but their causative role has not been confirmed. Even for genes whose mutations have been demonstrated to cause POI, many of their variants have not been connected with POI. Fourth, because of the heterogeneous nature of POI and the multiple modes of mutational spread, the prevalence and type of gene mutations are distinct among POI women of different ethnicities and different POI populations. Therefore, in the future, population stratification will be vital when analyzing genetic alterations. Fifth, the experimental results obtained from mouse models are not all applicable to humans because there are genetic and physiological differences between mice and humans. Moreover, the mouse models cannot deal with the extremely complex interactions among molecules, cells, organs, organisms, and the environment. Finally, according to many recent studies, digenic and oligogenic effects on POI have been observed, demonstrating that POI may not be a completely monogenic disease [40,72]. These are the questions that need to be addressed in the future.

In conclusion, the present review summarizes the genetic effects of POI in different fields (chromosomal variations, single gene mutations, impaired mitochondrial functions, and abnormal levels of non-coding RNAs). The findings clearly indicate that examining various genetic factors is crucial in determining the underlying etiologies of idiopathic POI cases. Therefore, this work summarizes and enriches our knowledge of POI etiology by providing the latest information concerning the selected genetic causes of POI obtained from patients and experimental animal models. However, this article is limited, as it only focuses on genetic causes.

## Figures and Tables

**Figure 1 ijms-24-04423-f001:**
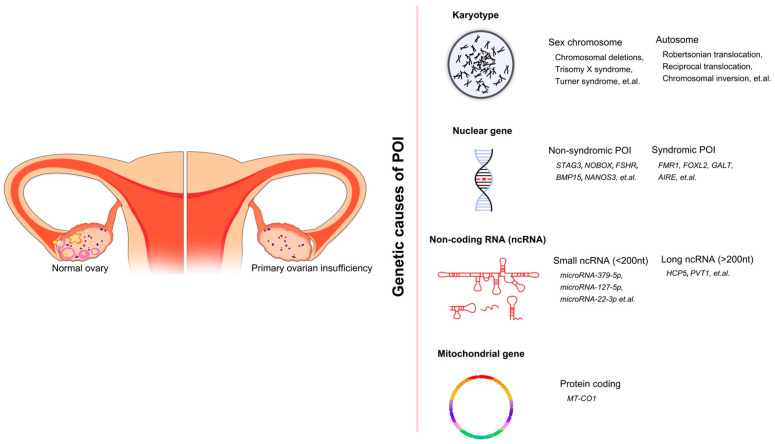
The overall genetic causes of POI, showing the conditions of ovaries in healthy women and POI women. Four categories of genetic factors and corresponding examples are included.

**Table 1 ijms-24-04423-t001:** List of candidate genes implicated in non-syndromic POI.

Classification	Gene	Mechanism of Action (MOA)	Evidence of POI-Relating
Meiosis and DNA Replication and Repair	HFM1	Homologous recombination and synapsis [38,39,40]	c.1006 + 1G > T, p.G1034S [41]
	PRIM1	DNA replication and repair [42]	SNP rs2277339 [43]
	STAG3	Synapsis [44]	c.1950C > A, p.Tyr650* [45]
	MCM8	Homologous recombination and DNA synthesis [46]	c.724T > C, p.C242R [47]
	MCM9	Homologous recombination and DNA synthesis [46]	c.1483G > T, p.E495* [48]
	DMC1	DNA strand exchange and DNA repair [49]	c.106G > A, p.Asp36Asn [50]
	MCL-1	Mitosis–meiosis transition [51]	MCL-1 knockout mice show similar presentations to POI patients [52]
	MSH4	Synapsis and DNA repair [38]	c.2261C > T, p.Ser754Leu [53]
	MSH5	Synapsis and DNA repair [38]	c. C1051G; p.R351G [54]
	MEIOB	Homologous recombination and DNA repairing [55]	c.258_259del [56]
	PSMC3IP	Homologous recombination [57]	c.489 C.G, p.Tyr163Ter [58]
	FANC group genes	Meiosis and folliculogenesis [59]	Mutations in FANC group genes can cause impaired meiosis and folliculogenesis such as c.1048_1051delGTCT, p.Gln350Valfs*18 in FANCL [60]
	SYCP2L	Synaptonemal complex assembly [61]	c.150_151del, p.Ser52profs*7 [61]
	HSF2BP	Homologous recombination [62]	c.382T > C, p.C128R [63]
	ZSWIM7	Homologous recombination [64]	c.173C > G, p.Ser58* [64]
Transcription Factor	SF-1	Gonadal and adrenal development and ovarian folliculogenesis and steroidogenesis [7]	c.74A > G, p.Y25C [65]
	NOBOX	Early folliculogenesis and cell cycle regulation [66,67]	c.567delG, p.T190Hfs*13 [67]
	SOHLH1	Gonadal glands and primordial follicles development [68]	p.Ser317Phe [69]
	SOHLH2	Gonadal glands and primordial follicles development [68]	c.235G > A, p.Glu79Lys [70]
	LHX8	Early oogenesis [71]	c.974C > T, p.A325V [72]
	FIGLA	Oogenesis [71]	c.2 T > C, p.Met1Thr [73]
Signaling molecules and receptors	FSHR	Follicular development and ovarian hormone regulation [74]	c.1253T > G, p.Ile418Ser [75]
	Inhibin family	Follicle-stimulating hormone regulation and folliculogenesis [33]	c.769G > A [76]
	AMHR2	Follicular development [77]	p.I209N [78]
	BMP15	Follicular development and granulosa cell protection [1]	c.791G > A, p.R264Q [79]
	GDF9	Follicular maturation and granulosa cell proliferation [80]	c.604C > T, p.Gln202* [81]
	LAT	Survival of granulosa cell [82]	c.245C > T and c.181C > G [82]
	VEGFA	Follicular development and ovarian function maintaining [83]	−1154G > A and 936C > T [83]
	BMPR1A	Formation of primordial follicle pool [84]	c.1325G > A, p.Arg442His [84]
	BMPR1B	Regulation of cumulus cell expansion and ovulation cycle [84]	c.761G > A, p.Arg254His [84]
RNA Metabolism and Translation	NANOS3	Cell antiapoptosis and translation repression [85]	c.358G > A, p.Glu120Lys [85]
	EIF4ENIF1	Translation repression [86]	c. 1286C > G, p.Ser429X [87]

**Table 2 ijms-24-04423-t002:** List of genes associated with POI (initiated via mitochondrial dysfunction) and their MOA.

Gene	Gene Type	Mechanism of Action (MOA)	References
RMND1	Nuclear gene	Supports translation of the mitochondrial DNA-encoded peptides.	[219]
MT-CO1	Mitochondrial gene	Mutations in MT-CO1 gene cause low COX activity, which is responsible for reduced ATP production. Additionally, the low level of ATP can give rise to follicular depletion via over-activating mTOR.	[220]
MRPS22	Nuclear gene	It is important in encoding the small subunit (28S) of mitochondrial ribosome and ovarian development.	[221]
CLPB	Nuclear gene	It encodes a mitochondrial disaggregase, which functions as a protein folding regulator and prevents oocyte damage.	[222]
TFAM	Nuclear gene	It gives rise to a component of replisome machinery of mitochondria. Additionally, the TFAM protein is associated with mtDNA replication and expression.	[223]

## Data Availability

Not applicable.

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
