# Peer review of "Selected Genetic Factors Associated with Primary Ovarian Insufficiency"

_ijms, 2023, doi:10.3390/ijms24054423_

Round 1

Reviewer 1 Report

 I have no recommendations for adding or changing the text of the review.

 I received a number of answers for myself on some questions about defects in ovarian folliculogenesis. Thanks.

Author Response

Dear reviewer:

Thanks very much for taking your time to review this manuscript. 

Reviewer 2 Report

The presented paper is focusing on POI genetic background concluding so far identified and I found it very interesting. In my opinion paper deserves publication, although some important points need to be corrected, rephrased or explained. Here is a major criticism:

However, due to its heterogeneity and fewer genetic causes that have been proven to contribute to POI, many POI cases are idiopathic.

Comment :

Idiopathic cases suggesting inability to identify causes, and not excluding even a substantial portion of genetic causes. This sentence somehow is depreciating the fundamental idea of this paper

Female infertility means that a woman fails to be pregnant after a year of preparation or happens to an over 35-year-old woman with 6 months of preparation

Comment :

Please rephrase this sentence

…… which can lead to many harmful and even lethal human genetic diseases such as Down syndrome and Turner syndrome

Comment :

Lethality if occurs, refers to early embryonal stages (miscarriages) and therefore not to Down or Turner syndromes, but to trisomies 21, 18, 13 and sex chromosomes rearrangements.

Specifically, all X monosomy and trisomy,  ………..

Specifically, all X monosomies and trisomies, (plural)

……… by complete or partial deletion of one X chromosome

What if  Y chromosome would be deleted ? Replace please X chromosome for sex chromosome

Trisomy X syndrome (TXS) with a karyotype 47XXX is another chromosomal numerical defect …..

Please replace all chromosomal numerical defect for aneuploidy or sex chromosome aneuploidy. That is correct and preferred clinical/cytogenetic entity.

Only 23 POI cases related to Robertsonian and reciprocal translocation and chromosomal inversion were identified in different POI populations with different ethnicities, such as Chinese, Thai and American [36]

Please rephrase

 ……… and 125 POI so that many genes such as BMP15, NOBOX and FMR1 have been discovered

Gene symbols should be written in italics –applies to the entire paper!

A recent research conducted WES in a Chinese POI pedigree with two POI patients (the proband and her mother) and found a novel missense mutation of  HFM1 [38].

Instead of using “novel mutation” or “heterozygous mutation” that is not really informative, provide a more detailed description of this change on protein level. The same plies to SNP accession numbers that are not providing much information about change (PRIM1 and so on

In female mice, loss of MCM8 and MCM9 contributes to sterility

Ortolog genes are not written in capitals!

Seven percent of POI patients in this cohort harbor DMC1 variants

What variants? Pathogenic VUS or benign?

Additionally, scientists identified five heterozygous 348 nonsynonymous mutations (deleterious or not)

(deleterious or not) –pathogenic VUS or benign?

FIGLA exists at 2p13.3

FIGLA is located at 2p13.3

With the help of the AGG triplet, the normal alleles contain about 5-44 repeats, but the  most frequent length exists in certain populations [174,175]

Please rephrase – unclear sentence. What does AGG triplet has to do with CGG expansion? What is the most frequent length? Please explain this dynamic mutation – what does a permutation means? Reader might be not familiar with this rare type of mutation.

Last, according to many recent studies, digenic and oligogenic effects on POI are also observed, pointing out that POI may not be a completely monogenetic disease [42,130]

Monogenic (or mendelian), not monogenetic

General suggestion: for providing mutation description on protein level, use consequently single or better tri-letter amino acid abbreviation (according to HGVS recommendation)

Reviewer 3 Report

A thorough review of known and potential causes of POI. While there is not novel data, this is a useful resouce. I recommend editing by a native English speaker for clarity. In addition, more on how this information may be utilized or next steps in further study or evaluation would elevate the review.

Reviewer 4 Report

Genetics or Primary Ovarian Insufficiency

Authors have written a review about the Genetic of Primary Ovarian Insufficiency (POI) in women. They discuss genetic factors and the proposed pathogenic mechanisms associated with them.

The manuscript divides genetic factors by: chromosomal abnormalities, single gene variants and non-syndromic POI, single gene mutations in syndromic and pleiotropic POI, mitochondrial dysfunction and abnormal levels of non-coding RNA. They have an extensive list of candidate genetic factors described in the manuscript.

The Introduction does not discuss other recent review papers on POI that discuss genetic factors, or build on what has already been published. What is novel about this work? Additionally the authors do not clear describe how they chose the list of genetic factors to discuss.

There is no Discussion section just a brief Conclusion. The reason for an increase in non – Turner POI in the population is not discussed. It is an increase in prevalence or better detection methods or both? What is the role of ethnicity? The role of the current diagnostic aids such as chromosomal, familial analysis, and genomic analysis and where this technology is headed should be better explained.

A companion table grouped by type of factor, name of factor, proposed mechanism of action, type of evidence for the non-syndromic POI.

The English was at times awkward, and many grammatical errors were present. In some sections there is a lack of clarity. A native English speaker should review the revised manuscript.

Specific comments:

Line 9 “genetic’ should not be capitalized

Line 24-25 Use of the term within a year of preparation is a non-medical term. Clarify.

Line 31 anti-Müllerian hormone? Ovarian reserve?

Line 37 reference?

Line 43 non-Turner’s POI, 4.6 per 100,000 person years a 10 fold lower prevalence in youth that estimated

Line 52 Cure a genetic defect???

Line 65 varies in

Line 78 rephrase original follicles?

Line 152 rephrase was first to demonstrate …. at the spindle pole

Line 204 rephrase  Not all of the aforementioned

Line 231 rephrase was MCL-1 had a higher level of expression after treatment

Line 232 – 238 needs to revised.  Cadmium is a heavy metal and is a known toxin with effects on the reproductive system. It has many effects on various cellular processes.

Line 236 rephrase authors in one report failed to find

Line 252 – 245  Rephrase scientists in these studies explored revise text here

Line 265 rephrase research group

Line 276 Don’t begin sentences with”And”

Line 282 Sweden….ethnicities

Line rephrase reproductive processes (not childbearing)

Line 303 is expressed in

Line 307 rephrase previous research study

Line rephrase in women … is mainly expressed in …., but the

Line 312-313 awkward…rephrase

Line 318 …is expressed in

Line 324 which is present in granulosa cells communicating with

Line 333 and these authors ? (researchers) confirmed

Line 339 ? super enhancer?

Line 350 ethnicity

Line 359 don’t begin sentences with And

Line 367 is needed

Line 369 is expressed

Line 374 ?gene called Zona Pellucida (accession number?)

Line 398 ethnicity

Line 420 susceptibility to

Line 449 and are expressed

Line 552 babies

Line 554 what is “enough primordial follicles”?

Line 567 rephrase testicular hormones??? This is unclear

Round 2

Reviewer 4 Report

Authors have written a review about the Genetics of Primary Ovarian Insufficiency (POI) in women.

Likely the title should be changed to Selected Genetic Factors Associated with Primary Ovarian Insufficiency.

They discuss genetic factors and the proposed pathogenic mechanisms associated with them of a selected number of genetic causes of POI.

Authors have addressed the grammatical concerns.

However a major concern is how the list of the various genetic orders was assembled from the last ten years for this paper....it is still not clear how the list was generated. More than 60 genes have been associated with POI. See doi: 10.12688/f1000research.26423.1

If another scientist tried in 5 years time wanted to find out what was new in the literature regarding genetic causes of POI how would they approach this? Generally they would use the same key words and delimit the years to search.

Typically in a Materials and Methods section a minimum statement of the key words used in a known high quality scientific search engine / data base are listed. As a reviewer I would be able to enter those key words and find the articles with the genes described in the manuscript. They would have to correspond to what was presented or an explanation of what was chosen would be and this case needs to be included. Another method is a  metaanalysis of the content of various research papers again using defined key words. Authors must state a reason for why and how you have selected the various genes in the manuscript.

The authors make a statement that has no bearing on the concern about how the genes were selected and the statement is unclear. "Moreover, we reviewed the genetic disorders by using PubMed and Google Scholar to screen out the genetic factors with a relatively high incidence and those that have been recently discovered in different populations."

Figure 1 shows a normal ovary on the left side and right side with the POI. A suggestion is to move the 4 categories to the right side of the uterine image to indicate visually they are associated with POI.

Line 89 correct term would be primordial follicle.

Line 269 begin sentence with Cadium not Cd

Line 357 change causative causes to underlying

Round 3

Reviewer 4 Report

Authors have written a review about the Genetics of Primary Ovarian Insufficiency (POI) in women

The manuscript has been revised

A few comments on the manuscript

The authors do not frame for the reader clearly how they chose the subset of genetic causes in their 4 categories.

Line 12 selected genetic causes

Line 19 remove the )

Figure 1 suggest the Genetic factors material is listed vertically rather than horizontally.

Line 72 Authors shall add additional detail to the description of the methodology

Line 72 The following key words were ….entered in Pubmed, list the string of the terms searched, an example would be “POI and genetic factors” describe the number of hits (articles),  “POI and genome” (number of articles) etc

Line 75 list the years searched, what languages were searched if applicable

List the total number of articles obtained using the search criteria. List the number of articles included in the review from the different sources (Pubmed vs Google Scholar). Described if peer review only articles and reviews were included. Other authors must be able to replicate your work.  

Line 75 rephrase …to be included in this review the selected publications had to focus on the following: identifying ….

Frame for the reader why the authors selected the included references.

Line 80 Thus, …. Remove this sentence.  

Line 370 Incomplete sentence underlying ? cause?

Line 809 add ...selected

Discuss why mouse models may not reflect what occurs in women.
